# Chinese Inertial GAN for Writing Signal Generation and Recognition

## Abstract

Disabled people constitute a significant part of the global population, deserving of inclusive consideration and empathetic support. However, the current human-computer interaction based on keyboards may not meet the requirements of disabled people. The small size, ease of wearing, and low cost of inertial sensors make inertial sensor-based writing recognition a promising human-computer interaction option for disabled people. However, accurate recognition relies on massive inertial signal samples, which are hard to collect for the Chinese context due to the vast number of characters. Therefore, we design a Chinese inertial generative adversarial network (CI-GAN) containing Chinese glyph encoding (CGE), forced optimal transport (FOT), and semantic relevance alignment (SRA) to acquire unlimited high-quality training samples. Unlike existing vectorization focusing on the meaning of Chinese characters, CGE represents the shape and stroke features, providing glyph guidance for GAN to generate writing signals. FOT constrains feature consistency between generated and real signals through the designed forced feature matching mechanism, meanwhile addressing GANs' mode collapse and mixing issues by introducing Wasserstein distance. SRA captures the semantic relevance between various Chinese glyphs and injects this information into the GAN to establish batch-level constraints and set higher standards of generated signal quality. By utilizing the massive training samples provided by CI-GAN, the performance of six widely used classifiers is improved from 6.7% to 98.4%, indicating that CI-GAN constructs a flexible and efficient data platform for Chinese inertial writing recognition. Furthermore, we release the first Chinese writing recognition dataset based on inertial sensors in GitHub.

## 1 Introduction

One of the most significant obstacles for disabled individuals in their daily lives is the lack of efficient human-computer interaction (HCI) methods [1]. Traditional keyboard-based HCI systems often fail to meet the specific needs of disabled users, particularly those who are visually impaired or have lost their fingers, which underscores the urgent need for developing technologies that cater to the unique requirements of disabled individuals [2]. Providing tailored HCI solutions not only enhances their quality of life and independence but also facilitates their integration into society, enabling greater participation in education, employment, and social activities. Such technological advancements hold profound significance, creating a more inclusive and equitable society.

As efficient motion-sensing components, inertial sensors can play a crucial role in recognizing writing movements. Inertial sensors can measure the acceleration and angular velocity of moving objects, making it possible to convert written characters into digital text [3, 4, 5, 6]. Due to their small size, ease of integration, low power consumption, and low cost, inertial sensors are widely used in electronic devices such as smartphones, smartwatches, and fitness bands [7, 8, 9, 10], making them particularly

suitable for disabled users. Inertial sensors can be integrated into wearable devices, providing a more accessible and user-friendly means for disabled individuals to interact with computers and other digital devices. By capturing the subtle movements of a user's hand or other body parts, inertial sensors can translate these motions into written text, enabling effective communication and interaction without the need for a traditional keyboard. In addition, unlike optical or acoustic sensors, inertial sensors are highly resistant to external factors such as lighting conditions, physical obstructions, or environmental noise, which showcases their unique robustness in motion capture [11, 12, 13, 14, 15]. Consequently, inertial sensors provide a medium for Chinese character writing recognition that aligns with natural writing habits and can be seamlessly integrated into the writing process. With the widespread adoption of smart devices, the technology of Chinese character writing recognition based on inertial sensors may redefine the Chinese character input in the digital age, offering disabled people a comfortable human-computer interaction methods.

However, the major challenge in achieving accurate Chinese writing recognition using inertial sensors is obtaining large-scale, diverse inertial writing data samples. For any recognition model aimed at accurately analyzing the complex strokes and structures of Chinese characters, it is crucial to train the model with extensive, diverse writing samples [16]. Considering that the collection and processing of Chinese writing samples are laborious and require high data quality and diversity, this task becomes exceedingly challenging and increasingly difficult as the number of characters increases. Therefore, generating realistic Chinese writing signals based on inertial sensors has become a central technological challenge in recognizing Chinese writing.

To acquire high-quality, diverse samples of inertial Chinese writing, we applied GAN for IMU writing signal generation for the first time and proposed CI-GAN, which can generate unlimited inertial writing signals for an input Chinese character, thereby providing rich training samples for Chinese writing recognition classifiers. CI-GAN provides a more intuitive and natural human-computer interaction method for the Chinese context and advances the application of smart devices with Chinese input. The main contributions of this paper are summarized as follows.

- Considering traditional Chinese character embedding methods that only focus on the meaning of characters, we propose a Chinese glyph encoding (CGE), which represents the shape and structure of Chinese characters. CGE not only injects glyph and writing semantics into the generation of inertial signals but also provides new tools for studying the evolution and development of hieroglyphs.

- We propose a forced optimal transport (FOT) loss for GAN, which not only avoids mode collapse and mode mixing during signal generation but also ensures feature consistency between the generated and real signals through a designed forced feature matching mechanism, thereby enhancing the authenticity of the generated signals.

- To inject batch-level character semantic correlations into GAN and establish macro constraints, we propose a semantic relevance alignment (SRA), which aligns the relevance between generated signals and corresponding Chinese glyphs, thereby ensuring that the motion characteristics of the generated signal conform to the Chinese character structure.

- Utilizing the training samples provided by CI-GAN, we increase the Chinese writing recognition performance of six widely used classifiers from 6.7% to 98.4%. Furthermore, we provide the application scenarios and strategies of 6 classifiers in writing recognition according to their performance metrics. For the sake of sharing, we release the first Chinese writing recognition dataset based on inertial sensors in GitHub.

## 2  Related Work

The technology for recognizing Chinese handwriting movements has the potential to bridge the gap between traditional writing and digital input, providing disabled individuals with a natural way of writing and greatly enhancing their ability to participate in digital communication, education, and employment. It also offers a new human-computer interaction avenue for normal people. Hence, Chinese handwriting movement recognition has garnered significant attention in recent years, leading to numerous related research achievements. Ren et al. utilized the Leap Motion device to propose an RNN-based method for recognizing Chinese characters written in the air [17]. The Leap Motion sensor, consisting of two infrared emitters and two cameras, can accurately capture the motion of hands in three-dimensional (3D) space [18]. However, the Leap Motion device is sensitive to lighting

conditions, and either too strong or too weak light can interfere with the transmission and reception of infrared rays, affecting the recognition effect [19]. Additionally, the detection space of the Leap Motion device is an inverted quadrangular pyramid, limiting its field of view. Movements outside this range cannot be captured. Most importantly, the Leap Motion device is expensive and requires a connection to a computer or VR headset to function, severely limiting its application prospects [20].

As wireless networks become more prevalent, Wi-Fi signals are gradually being applied to motion capture [21, 22]. Since Wi-Fi signals can penetrate objects and are unaffected by lighting conditions, they have a broader application scope than optical motion capture systems [23, 24]. Guo et al. used the channel state information (CSI), extracted from Wi-Fi signals reflected by hand movements, to recognize 26 air-written English letters [25]. However, while Wi-Fi signals do not have visual range limitations and can penetrate obstacles, they are easily disturbed by other signals on the same unlicensed band, severely affecting system performance. Moreover, the sampling frequency and resolution of Wi-Fi signals are very limited, making it difficult to capture detailed information during the writing process and, thus, hard to recognize air-written Chinese characters accurately [26, 27].

Despite the advantages of low cost, wearability, and low power consumption offered by inertial sensors, there is currently a lack of large-scale, high-quality public datasets, causing few studies to use inertial sensors for 3D Chinese handwriting recognition [28, 29, 30, 31]. To collect data, Zhang et al. employed 12 volunteers, each of whom was asked to write the assigned Chinese characters on paper 30 times [32]. The inertial measurement unit (IMU) built into smartwatches was used to collect the motion signals of the volunteers while writing, ultimately achieving a recognition accuracy of 90.2% for 200 Chinese characters. However, this study aims to identify the signals of normal individuals writing on paper, which is not applicable to people with disabilities. Moreover, this method can only realize desktop-based 2D writing recognition, which reduces the comfort and flexibility of the writing process, inherently limiting the application scenarios of Chinese handwriting recognition. Additionally, this method cannot effectively recognize massive Chinese characters due to the physical and mental limitations of volunteers for data collection. Considering the vast number of Chinese characters, providing large-scale, high-quality writing signal samples for each character is nearly impossible, which has become the most significant bottleneck limiting the development of Chinese handwriting recognition technology based on inertial sensors. Therefore, designing a model for generating Chinese handwriting signals provides researchers with an endless supply of signal samples and a flexible, convenient experimental data platform, accelerating the development and testing of new algorithms and supporting the research and application of Chinese handwriting recognition.

## 3 Method

To generate inertial writing signals for Chinese characters, we propose the Chinese inertial generative adversarial network (CI-GAN), as shown in Fig. 1. For an input Chinese character, its one-hot encoding is transformed into glyph encoding using our designed glyph encoding dictionary, which stores the glyph shapes and stroke features of different Chinese characters. Thus, the obtained Chinese glyph encoding contains rich writing features of the input character. This glyph encoding, along with a random noise vector, is fed into a GAN, generating the synthetic IMU signal for the character, where glyph encoding provides glyph and stroke features of the input character, while the random noise introduces randomness to the virtual signal generation, ensuring the diversity and variability of the generated signals. To ensure that the GAN learns the IMU signal patterns for each character, we designed a forced optimal transport (FOT) loss, which not only mitigates the issues of mode collapse and mode mixing typically observed in GAN frameworks but also forces the generated IMU signals to closely resemble the actual handwriting signals in terms of semantic features, fluctuation trends, and kinematic properties. Moreover, a semantic relevance alignment (SRA) is proposed to provide batch-level macro constraints for GAN, thereby keeping the correlation between generated signals consistent with the correlation between Chinese character glyphs. Equipped with CGE, FOT and SRA, CI-GAN can provide unlimited high-quality training samples for Chinese character writing recognition, thereby enhancing the accuracy and robustness of various classifiers.

### 3.1 Chinese Glyph Encoding

In one-hot encoding, each Chinese character is represented by a high-dimensional sparse vector (where only one element is 1, and all others are 0), which results in all characters being equidistant in the vector space, thereby losing the abundant semantic information contained in the characters.

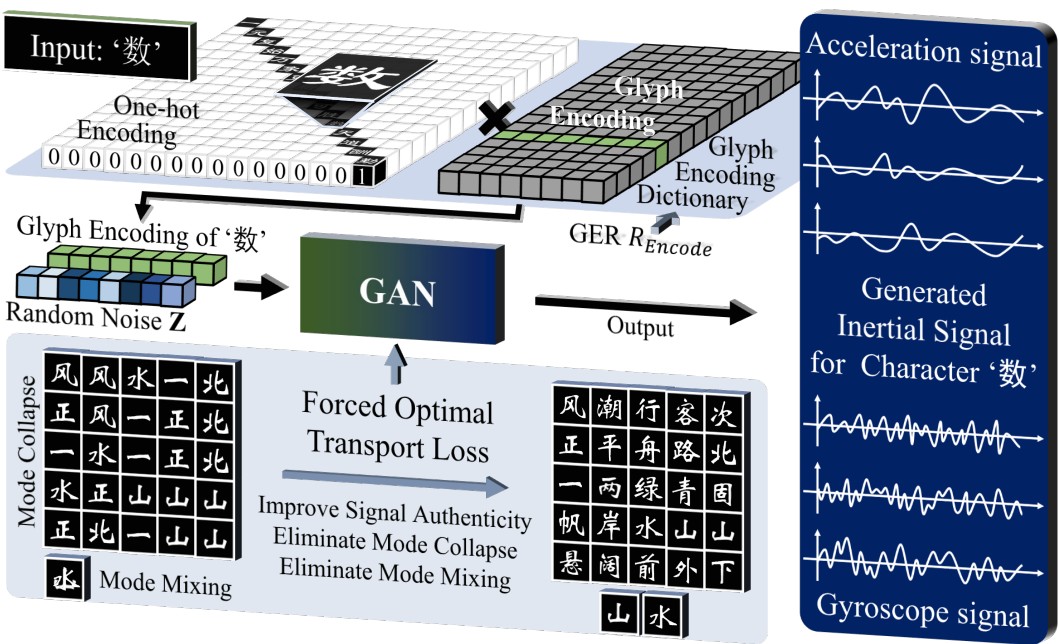

Figure 1: Flowchart of Chinese inertial generative adversarial network. The Chinese character "数" is input into the model, and its one-hot encoding is converted into glyph encoding (green cubes), which is then input into GAN together with random noise (blue cubes of different colors).

Therefore, one-hot encoding fails to inject rich information into GAN. Although there are some commonly used Chinese character embeddings, these embeddings store meaning information of the characters, not glyph information (i.e., shape, structure and writing strokes). For example, the characters "天" (sky) and "夫" (husband) are quite similar in writing motions, but their meanings are significantly different. To this end, we propose a Chinese glyph encoding (CGE), which encodes Chinese characters based on their glyph shapes and writing actions.

Considering that the inertial sensor signals capture the writing motion of Chinese characters, the motion signal exactly contains glyph information, which encourages simultaneous learning signal generation and Chinese glyph encoding under the supervision of real signals. Therefore, we create a learnable weight matrix $W$ after the one-hot input layer to capture the glyph information. When a Chinese character is input into CI-GAN in one-hot encoding, it first passes through this weight matrix. Since only one element in the one-hot encoding is 1, and the rest are 0, multiplying one-hot encoding by the weight matrix $W$ means obtaining one row of the matrix $W$. Hence, each row of $W$ can be seen as an encoding of a Chinese character, and this matrix can serve as a glyph encoding dictionary of Chinese characters. However, an unguided Chinese encoding dictionary often struggles to capture the differences in glyph shapes among different characters, assigning similar glyph encodings to characters with distinct glyphs. To address this, we propose a glyph encoding regularization (GER), which enhances the orthogonality of all character encoding vectors and increases their information entropy to store as many glyph features of the characters as possible, thereby avoiding triviality like one-hot encoding. Specifically, we use the $\alpha$-order Rényi entropy to measure the information content of the glyph encoding dictionary $W$, calculated as follows:

$$S_\alpha(W) = \frac{1}{1-\alpha}\log_2(tr(\tilde{G}^\alpha)), \text{where } \tilde{G}_{ij} = \frac{1}{N}\frac{G_{ij}}{\sqrt{G_{ii} \cdot G_{jj}}}, G_{ij} = \left\langle W^{(i)}, W^{(j)} \right\rangle. \quad (1)$$

where, $N$ represents the number of Chinese characters, which corresponds to the number of rows in the weight (encoding) matrix $W$. $G$ is the Gram matrix of $W$, where $G_{ij}$ equal to the inner product of the $i$-th and $j$-th rows of $W$, and $\tilde{G}$ is the trace-normalized $G$, i.e., $tr(\tilde{G}) = 1$. In similar problems, $\alpha$ is generally set to 2 for optimal results. $S_\alpha(W)$ measures the information content of the glyph encoding matrix $W$. A larger $S_\alpha(W)$ indicates more information encoded in $W$, meaning the glyph encodings are more informative. Meanwhile, as $S_\alpha(W)$ increases, all elements in the Gram matrix $G$ are forced to decrease, indicating that different encoding vectors have stronger orthogonality. It

is evident that the improvement of $S_\alpha(W)$ simultaneously enhances the information content and the orthogonality among the encodings. In light of this, the glyph encoding regularization $R_{\text{encode}}$ is constructed as $R_{\text{encode}} = \frac{1}{S_\alpha(W)}$. As $R_{\text{encode}}$ decreases during training, $S_\alpha(W)$ gradually increases, meaning the glyph encoding dictionary stores more information while enhancing the orthogonality among all Chinese glyph encodings, effectively representing the differences in glyph shapes among all characters. Thus, this glyph encoding can inject sufficient glyph information into GAN, ensuring that the generated signals maintain consistency with the target character's glyph.

## 3.2 Forced Optimal Transport

Ensuring the authenticity of virtual signals poses the greatest challenge when generating diverse signals, especially in following physical laws and simulating the potential dynamical characteristics of actual motions. To this end, we propose the forced feature matching (FFM), which ensures that the generated signal feature closely matches the real signal feature and the corresponding glyph encoding. Specifically, we use a pre-trained variational autoencoder to extract the real signal feature $h_T$ and generated signal feature $h_G$. Then, the consistency of $h_T$, $h_G$, and the corresponding glyph encoding $e$ is constrained by $\mathcal{L}_{FFM}$.

$$\mathcal{L}_{FFM} = 1 - \frac{\langle h_G, h_T \rangle + \langle h_G, e \rangle + \langle e, h_T \rangle}{\|h_G\| \|h_T\| + \|h_G\| \|e\| + \|e\| \|h_T\|}. \tag{2}$$

Another critical challenge lies in the mode collapse and mode mixing issue inherent to GAN architectures. Mode collapse limits the diversity of generated signal samples, causing GAN to generate signals only for a few Chinese characters, regardless of the diversity of input. On the other hand, mode mixing problems cause the generated signal to contain blend characteristics of multiple modes, which is unrealistic and unrecognizable. To address these issues, we introduce the optimal transport to GAN, which utilizes the Wasserstein distance as a loss function. Traditional GANs use the Jensen-Shannon divergence as the loss metric, which becomes ineffective when the distributions of real and generated data have little overlap, leading to mode collapse. The Wasserstein distance provides a more effective gradient even when the distributions are disjoint or significantly different, thereby preventing mode collapse. Furthermore, unlike the Jensen-Shannon divergence, the Wasserstein distance exhibits insensitivity to the balance between the training of the generator and discriminator, thereby alleviating mode mixing (We provide a rigorous mathematical proof in Appendix C). Combing OT and FFM constraints, we can obtain the forced optimal transport loss $\mathcal{L}_{FOT} = W(\mathbb{P}_T, \mathbb{P}_G) + \lambda \cdot \mathcal{L}_{FFM}$, where $W(\mathbb{P}_T, \mathbb{P}_G)$ is the optimal transport loss, representing the Wasserstein distance between the distributions of real and generated signals, enhancing the stability and diversity of the samples. $\lambda$ is a weighting coefficient for the forced feature matching loss $\mathcal{L}_{FFM}$. As $\mathcal{L}_{FFM}$ decreases during training, the generated signals increasingly approximate the characteristics of real signals.

## 3.3 Semantic Relevance Alignment

As motion records of Chinese writing, the semantic relationships between generated signals should align with the relationships between Chinese character glyphs. To ensure the generated inertial signals accurately reflect the character relationships between Chinese character glyphs, we propose semantic relevance alignment (SRA), which ensures consistency between the glyph encoding relationships and the signal feature relationships, thereby providing batch-level macro guidance for GANs and enhancing the quality of the generated signals. For each batch of input Chinese characters, we compute the pairwise cosine similarities of their Chinese glyph encodings to form an encoding similarity matrix $M_e$. Simultaneously, the pairwise cosine

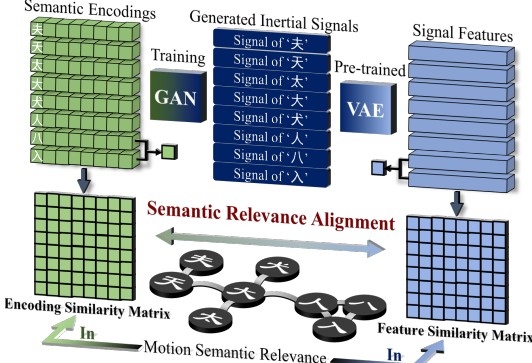

Figure 2: Diagram of semantic relevance alignment.

similarities of generated signal features (extracted by the pre-trained VAE) are computed to form a feature similarity matrix $M_h$. Then, the loss of semantic relevance alignment $\mathcal{L}_{SRA} = \|M_h - M_e\|_2^2$ is established to minimize the difference between the two matrices, thereby ensuring that the semantic relationships in the input character glyphs are accurately contained in the generated signals.

## 4 Experiments and Results

### 4.1 Data Collection and Experimental Setup

We invited nine volunteers, each using their smartphone's built-in inertial sensors to record handwriting movements. The nine smartphones and their corresponding sensor models are listed in Table 1. Each volunteer held their phone according to their personal habit and wrote 500 Chinese characters in the air (sourced from the "Commonly Used Chinese Characters List" published by the National Language Working Committee and the Ministry of Education), writing each character only once. In total, we obtained 4500 samples of Chinese handwriting signals. We randomly selected 1500 samples from three volunteers as the training set, while the remaining 3000 samples from six volunteers were used as the test set without participating in any training. All experiments are implemented by Pytorch 1.12.1 with an Nvidia RTX 2080TI GPU and Intel(R) Xeon(R) W-2133 CPU.

Table 1: The built-in IMU specifications of some smartphones. Note that since the IMUs in some types of iPhones are customized by the manufacturer, the model and price are not disclosed.

| Dataset | Smartphone | Release Time | IMU | Unit price |
|---|---|---|---|---|
| Training | iPhone 13 pro | Sep. 2021 | Undisclosed | / |
| | HUAWEI P40 | Mar. 2020 | LSM6DSM | $0.30 |
| | HUAWEI P40 Pro | Apr. 2020 | LSM6DSO | $0.33 |
| Testing | iPhone 14 | Sep. 2022 | Undisclosed | / |
| | iPhone 15 | Sep. 2023 | Undisclosed | / |
| | VIVO T2x | May. 2022 | LSM6DSO | $0.33 |
| | OPPO Reno 6 | May. 2021 | ICM-40607 | $0.28 |
| | Realme GT | Mar. 2021 | BMI160 | $0.21 |
| | Redmi K40 | Mar. 2021 | ICM-40607 | $0.28 |

### 4.2 Signal Generation Visualization

To visually demonstrate the signal generation effect of CI-GAN, we visualized the real and generated inertial sensor signals of the handwriting movements for the Chinese characters "科" and "学", respectively. In these figures, the blue curves represent the three-axis acceleration signals, and the yellow curves represent the three-axis gyroscope signals. It can be observed that the generated signals closely follow the overall fluctuation trends of the real signals, indicating that CI-GAN effectively preserves the handwriting movement information of the real signals. To further verify the consistency

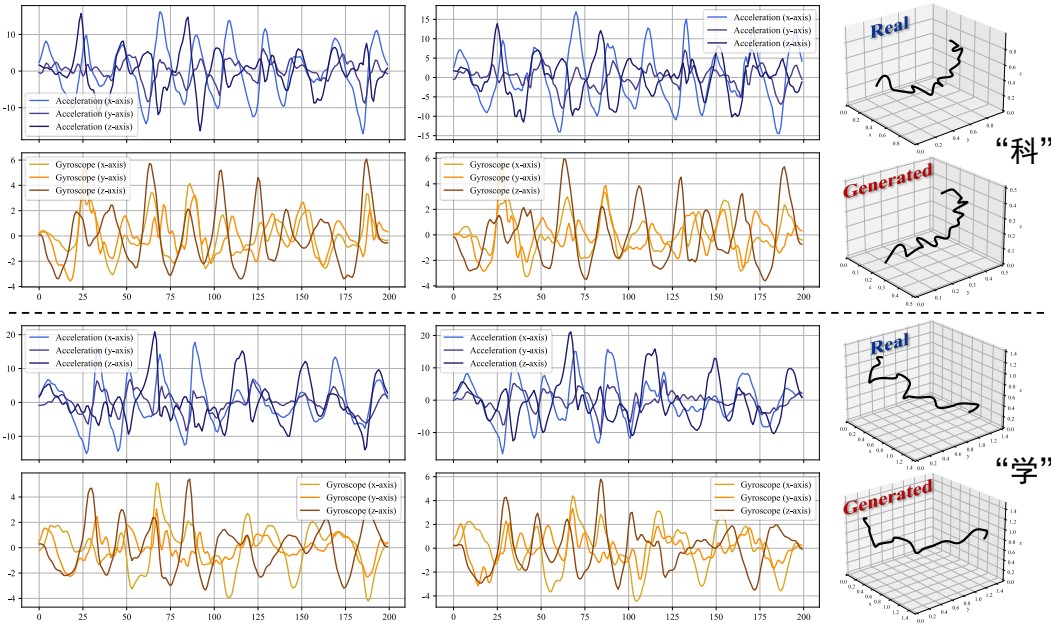

Figure 3: The visualization results of the 6-axis signals recorded by the inertial sensor for different Chinese character writing movements and the corresponding generated signals. The left side is the original inertial sensor signal, the middle is the corresponding generated signal, and the right side is the reconstructed writing trajectory.

of the movement characteristics between the generated and real signals, we employed a classical inertial navigation method [33] to convert both the real and generated signals into corresponding

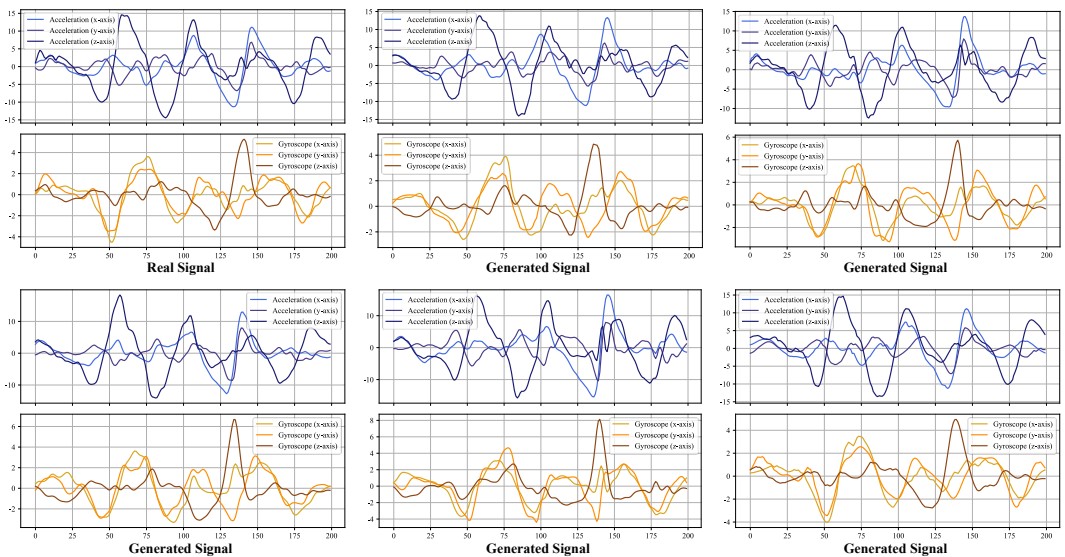

Figure 4: Visualization of the real IMU signal for writing "王" and the virtual signals generated by CI-GAN. The upper left corner is the real signal, and the remaining signals are virtual signals.

motion trajectories, as shown in the third column of Fig. 3. It is important to note that the purpose of reconstructing the motion trajectories is not to precisely reproduce every detail of the writing process but to compare the overall shape similarity between the trajectories derived from real and generated signals. The highly similar shapes between the trajectories indicate that the generated signals accurately capture the structural information of different Chinese characters and can effectively simulate the key movement features of the handwriting process, including stroke order, movement direction changes, and velocity variations. Additionally, the obvious differences in details between the real and generated signals demonstrate CI-GAN's capability to generate diverse signals. Since the generated signals maintain the core movement and semantic features of the handwriting process, these differences do not impair the overall recognition of the characters but rather enhance the diversity of the training data.

To demonstrate CI-GAN's ability to generate unlimited high-quality signals, we generated five IMU handwriting signals for the same character "王" and compared them with a real handwriting signal, as shown in Fig. 4. We chose this character because its strokes are distinctly separated, making it easier to compare the consistency of stroke features between the generated and real signals. It can be observed that the generated signals exhibit similar fluctuation patterns to the real signal in all three axes of acceleration and gyroscope measurements, verifying CI-GAN's precision in capturing dynamic handwriting characteristics. Although the overall trends of the generated signals align with the real signal, the individual features show variations, demonstrating CI-GAN's potential to produce large-scale, high-quality, and diverse IMU handwriting signal samples.

## 4.3 Comparative Experiments

Using the trained CI-GAN, we generated 30 virtual IMU handwriting signals for each character, resulting in a total of 16500 training samples. To evaluate the impact of the generated signals on handwriting recognition tasks, we trained six representative time-series classification models with these training samples: 1DCNN, LSTM, Transformer, SVM, XGBoost, and Random Forest (RF). We then tested the performance of these classifiers on the test set, as shown in Fig. 5.

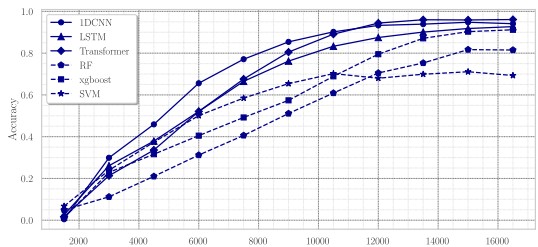

Figure 5: The recognition accuracy of 6 classifiers with varied training samples provided by CI-GAN.

When the number of training samples is small (1500 real samples), the recognition accuracy of all classifiers is poor, with the highest accuracy being only 6.7%. As the generated training samples are

introduced, all classifiers' recognition accuracy improves significantly, whereas deep learning ones such as 1DCNN, LSTM, and Transformer show the most notable improvement. When the number of training samples reaches 15000, the recognition accuracy of 1DCNN can reach 95.7%, improving from 0.87% (without data augmentation). The Transformer captures long-range dependencies in time-series data through its self-attention mechanism, enabling it to understand complex movement patterns. However, its excellent recognition ability relies on large amounts of data, making its performance improvement the most significant as CI-GAN continuously generates training data, improving from 1.7% to 98.4%. Compared to deep learning models, machine learning models also exhibit significant dependence on the amount of training data, highlighting the critical role of sufficient generated signals in handwriting recognition tasks. With the abundant training samples generated by CI-GAN, six classifiers achieve accurate recognition even for similar characters as shown in Appendix A.1.

In summary, CI-GAN provides a data experimental platform for Chinese writing recognition, enabling various classifiers to utilize the generated samples for training and improving their

Table 2: Performance comparison of 6 classfiers.

| Classifier | 1DCNN | LSTM | Transformer | RF | XGBoost | SVM |
|---|---|---|---|---|---|---|
| Runtime (s) | 0.00743 | 0.13009 | 0.03439 | 0.01269 | 0.00154 | 0.00173 |
| Memory (MB) | 22.153 | 29.897 | 52.336 | 35.418 | 19.472 | 3.881 |
| Accuracy | 95.7% | 93.9% | 98.4% | 83.5% | 93.1% | 74.6% |

recognition accuracy. To help researchers select suitable classifiers for different application scenarios, we further tested the recognition speed and memory usage of different classifiers for a single input sample and summarized their recognition accuracy in Table 2. Among the three deep learning models, 1DCNN has the fastest runtime and the smallest memory usage, with a recognition accuracy of 95.7%, slightly lower than the Transformer but sufficient for most practical applications. It is more suitable for integration into memory and computation resource-limited smart wearable devices such as phones, watches, and wristbands. In contrast, Transformer has the highest accuracy among the six classifiers and the highest memory usage, making it more suitable for PC-based applications. Compared to deep learning classifiers, traditional machine learning classifiers generally have lower accuracy, but with the support of abundant training samples generated by CI-GAN, the XGBoost model still achieves a recognition accuracy of 93.1%, very close to deep learning classifiers. More importantly, XGBoost, as a tree model, has strong interpretability, allowing users to intuitively observe which features significantly impact the model's decision-making process, which is a strength that deep learning models lack. Additionally, XGBoost's runtime and memory usage are better than the three deep learning classifiers, making it outstanding in scenarios requiring a balance between model performance, interpretability, and resource efficiency. For example, XGBoost can be integrated into stationery and educational tools to analyze students' handwriting habits and provide personalized feedback suggestions. Similarly, in the healthcare field, XGBoost can be used to analyze patients' writing characteristics, assisting doctors in evaluating treatment effects or predicting disease risks. Its high interpretability can provide an auxiliary reference for medical decisions and treatment plans, increasing patients' trust in the treatment.

## 4.4 Ablation Study

Systematic ablation experiments are conducted to evaluate the contributions of the CGE, FOT, and SRA modules in CI-GAN. We generated writing samples using the ablated models and trained the six classifiers on these samples. The results are summarized in Table 3. When no generated data is used (No augmentation), the recognition accuracy of all classifiers is very poor. Employing the Base GAN to

Table 3: Performance comparison of six classifiers trained on samples generated by different ablation models.

| Ablation model | 1DCNN | LSTM | Transformer | RF | XGBoost | SVM |
|---|---|---|---|---|---|---|
| No augmentation | 0.87% | 2.6% | 1.7% | 4.9% | 1.2% | 6.7% |
| w/o all (Base GAN) | 18.5% | 14.8% | 15.7% | 12.4% | 20.5% | 8.4% |
| w/ OT | 26.4% | 28.6% | 27.3% | 21.0% | 30.9% | 20.9% |
| w/ FOT | 39.9% | 38.0% | 35.3% | 31.9% | 46.8% | 27.3% |
| w/ CGE | 54.6% | 51.2% | 47.9% | 38.6% | 57.5% | 34.1% |
| w/ FOT+CGE | 80.7% | 80.5% | 80.9% | 57.2% | 70.4% | 59.5% |
| w/ FOT+CGE+SRA (CI-GAN) | 95.7% | 93.9% | 98.4% | 83.5% | 93.1% | 74.6% |

generate training samples brings slight improvement but still underperforms, underscoring the critical importance and necessity of data augmentation for accurate recognition. This also indicates that utilizing GAN to improve classifier performance is a challenging task. Introducing CGE, FOT, and SRA individually into the GAN significantly improves its performance, with the introduction of CGE bringing the most noticeable improvement. This demonstrates that incorporating Chinese glyph encoding into the generative model is crucial for accurately generating writing signals. When CGE, FOT, and SRA are simultaneously integrated into the GAN (i.e., CI-GAN), the performance of all six classifiers is improved to above 70%, with four classifiers achieving recognition accuracies exceeding

90%. Notably, the Transformer classifier achieves an impressive accuracy of 98.4%. Furthermore, statistical significance analysis is performed to validate the reliability of these results, as shown in Appendix A.2.

### 4.5 Visualization Analysis of Chinese Glyph Encoding

To demonstrate the effectiveness of the Chinese glyph encoding in capturing the glyph features of Chinese characters, we conducted a visualization analysis using t-SNE, which reduced the dimensionality of the glyph encodings of 500 Chinese characters and visualized the results in a 2D space, as shown in Fig. 6, where each point represents a Chinese character. For the convenience of observation, we selected 6 local visualization regions from left to right and zoomed in on them at the bottom. It can be observed that characters with similar strokes and structure (e.g., "办-为", "目-且", "人-入-八") are close to each other. Additionally, the figure shows several clusters where characters within the same cluster share similar radicals, structures, or strokes, indicating that CGE effectively captures the similarities and differences in the glyph features of Chinese characters. By incorporating CGE into

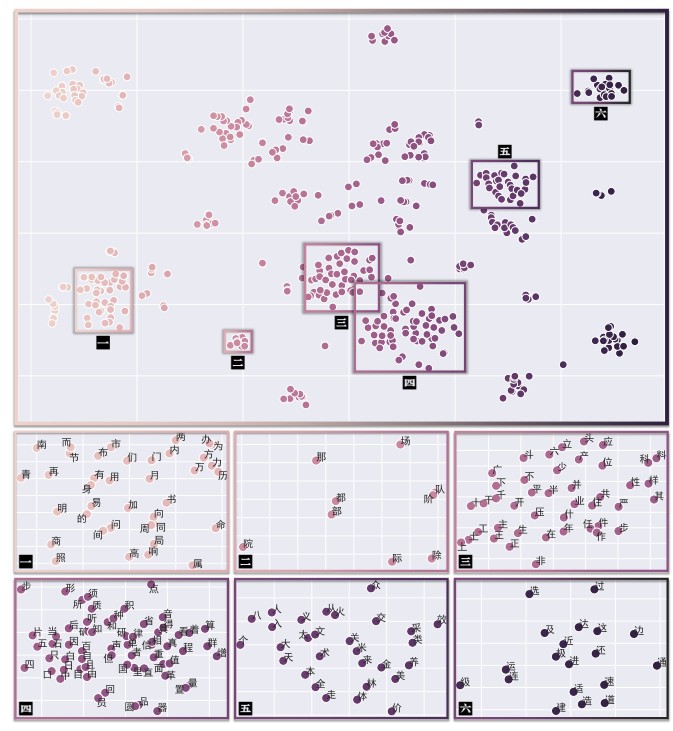

Figure 6: The t-SNE visualization of Chinese glyph encodings.

the generative model, CI-GAN can produce writing signals that accurately reflect the structure and stroke features of Chinese characters, ensuring the generated signals closely align with real writing movements. This encoding is not only crucial for guiding GANs in generating writing signals but also potentially provides new tools and perspectives for studying the evolution of Chinese hieroglyphs.

## 5 Conclusion

This paper introduces GAN to generate inertial sensor signals and proposes CI-GAN for Chinese writing data augmentation, which consists of CGE, FOT, and SRA. The CGE module constructs an encoding of the stroke and structure for Chinese characters, providing glyph information for GAN to generate writing signals. FOT overcomes the mode collapse and mode mixing problems of traditional GANs and ensures the authenticity of the generated samples through a forced feature matching mechanism. The SRA module aligns the semantic relationships between the generated signals and the corresponding Chinese characters, thereby imposing a batch-level constraint on GAN. Utilizing the large-scale, high-quality synthetic IMU writing signals provided by CI-GAN, the recognition accuracy of six widely used classifiers for Chinese writing recognition was improved from 6.7% to 98.4%, which demonstrates that CI-GAN has the potential to become a flexible and efficient data generation platform in the field of Chinese writing recognition. This research provides a novel human-computer interaction, especially for disabled people. Its limitations and impact are discussed in Appendix B.1 and B.2. In the future, we plan to extend CI-GAN to generate signals from other modalities of sensors, constructing a multimodal human-computer interaction system tailored for disabled individuals, which can adapt to the diverse needs of users with different disabilities. Through continuous collaboration with healthcare professionals and the disabled community, we will refine and optimize these multimodal systems to ensure they deliver the highest functionality and user satisfaction. Ultimately, this research aims to foster a society where digital accessibility is a fundamental right, ensuring that all individuals, regardless of physical abilities, can engage fully and independently with the digital world.

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

# Appendix / Supplemental Material

## A    Additional Experimental Results

### A.1    Performance of Classifiers on Similar Characters

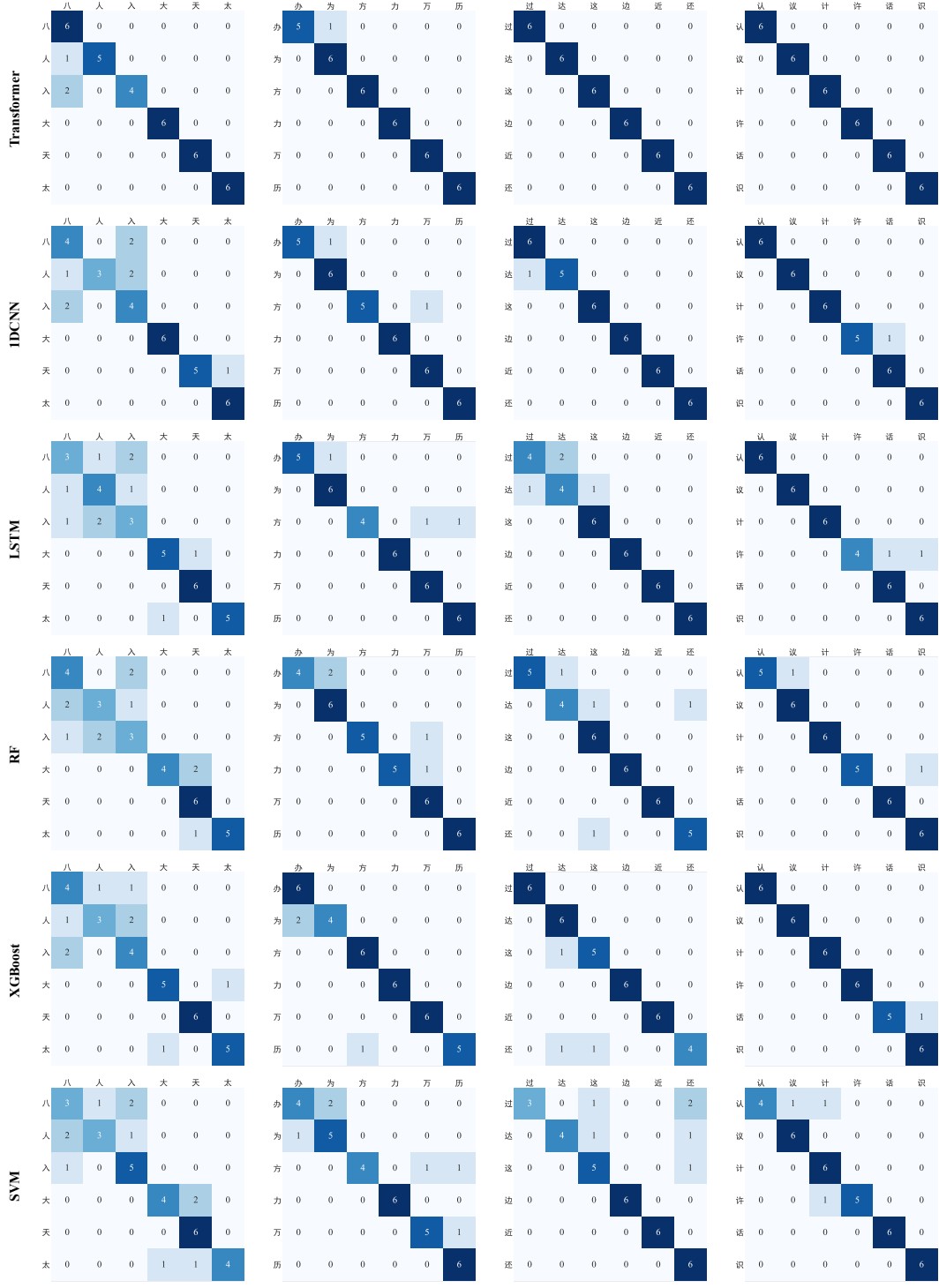

Figure 7: Confusion matrices of different classifiers for recognition results of Chinese characters with similar glyphs.

With the abundant training samples generated by CI-GAN, the handwriting recognition performance of all six classifiers significantly improved. To further verify the recognition performance of different classifiers on characters with similar strokes and glyphs, we selected four groups of characters with similar handwriting movements from the test set ("八人入大天太", "办为方力万历", "过达这边近还", and "认议计许话识") and presented the recognition results of the six classifiers in confusion matrices, as shown in Fig. 7. It can be observed that the values on the diagonal of all confusion matrices are significantly higher than the non-diagonal values, indicating high recognition accuracy for these similar handwriting characters with the help of samples generated by CI-GAN. However, some characters are still misrecognized. For instance, the characters "八", "人", and "入" have extremely similar structures and writing movements, posing challenges even when massive training samples are provided. Moreover, continuous and non-standard writing can also cause recognition obstacles. For instance, although the characters "过" and "达" have different strokes in static form, they are very similar in dynamic handwriting. Despite these challenges, the synthetic IMU handwriting samples generated by CI-GAN significantly enhance the classifiers' ability to recognize characters with similar glyph structures and handwriting movements, highlighting the value and significance of the proposed CI-GAN method. By providing diverse and high-quality training samples, CI-GAN improves handwriting recognition classifiers' performance and generalization ability, making it a valuable tool for advancing Chinese handwriting recognition technology.

## A.2 Statistical Significance Analysis

The CI-GAN model demonstrates significant performance improvements across multiple classifiers, as shown in Table 4. The Transformer classifier, for instance, achieves a mean accuracy of 98.4%, compared to 15.7% with the traditional GAN and 1.7% without data augmentation. This highlights CI-GAN's ability to generate realistic and diverse training samples that enhance handwriting recognition. Moreover, CI-GAN consistently improves accuracy and stability for all classifiers tested. The 1DCNN's accuracy increases to 95.7% from 18.5% with the traditional GAN and 0.87% without augmentation. Similarly, other models, including LSTM, RandomForest, XGBoost, and SVM, show substantial gains, underscoring CI-GAN's effectiveness across diverse machine-learning contexts. In addition, the narrow 95% confidence intervals, such as [98.2822%, 98.5178%] for the Transformer, validate the statistical significance and reliability of these results. This confirms CI-GAN's potential to consistently enhance classifier performance. In conclusion, CI-GAN represents a major advancement in Chinese handwriting recognition by generating high-quality, diverse inertial signals. This significantly boosts the accuracy and reliability of various classifiers, demonstrating CI-GAN's transformative potential in the field.

Table 4: Performance of different classifiers with CI-GAN generated data

| Ablation | Classifier | Mean Accuracy | Standard Deviation | 95% Confidence Interval |
|---|---|---|---|---|
| No data augmentation | 1DCNN | 0.87% | 0.11% | [0.8018%, 0.9382%] |
| | LSTM | 2.61% | 0.20% | [2.4761%, 2.7239%] |
| | Transformer | 1.70% | 0.13% | [1.6194%, 1.7806%] |
| | RandomForest | 4.89% | 0.09% | [4.8439%, 4.9556%] |
| | XGBoost | 1.20% | 0.15% | [1.1071%, 1.2929%] |
| | SVM | 6.65% | 0.10% | [6.5881%, 6.7119%] |
| Traditional GAN | 1DCNN | 18.5% | 0.16% | [18.4008%, 18.5992%] |
| | LSTM | 14.8% | 0.37% | [14.5707%, 15.0293%] |
| | Transformer | 15.7% | 0.15% | [15.6071%, 15.7929%] |
| | RandomForest | 12.4% | 0.17% | [12.2948%, 12.5052%] |
| | XGBoost | 20.5% | 0.23% | [20.3573%, 20.6427%] |
| | SVM | 8.40% | 0.34% | [8.1893%, 8.6107%] |
| CI-GAN | 1DCNN | 95.7% | 0.24% | [95.5513%, 95.8487%] |
| | LSTM | 93.9% | 0.53% | [93.5713%, 94.2287%] |
| | Transformer | 98.4% | 0.19% | [98.2822%, 98.5178%] |
| | RandomForest | 83.5% | 0.35% | [83.2831%, 83.7169%] |
| | XGBoost | 93.1% | 0.46% | [92.8148%, 93.3852%] |
| | SVM | 74.6% | 0.38% | [74.3644%, 74.8356%] |

## B  Discussion

### B.1  Societal Impact

CI-GAN model significantly improves the accuracy of Chinese writing recognition and offers an alternative means of human-computer interaction that can overcome the limitations of traditional keyboard-based methods, which are often inaccessible to those who are blind or lose their fingers. By providing a more accessible and user-friendly way to interact with digital devices, inertial sensors can facilitate effective communication, enhance the participation of disabled people in education and employment, and promote greater independence. Moreover, by addressing the unique needs of this population, such technological advancements reflect a commitment to inclusivity and social justice, ensuring that everyone, regardless of their physical abilities, has the opportunity to fully participate in and contribute to society.

Furthermore, by releasing the world's first Chinese handwriting recognition dataset based on inertial sensors, this research provides valuable data resources for both academia and industry, facilitating further studies and advancements. Additionally, the technology offers an intuitive and efficient learning tool for Chinese language learners, aiding in preserving and disseminating Chinese cultural heritage and strengthening the global influence of Chinese characters. In summary, the CI-GAN technology achieves not only significant breakthroughs in algorithmic research but also demonstrates extensive practical potential and substantial societal value, thereby being adopted by educational aid device manufacturers. This study provides a solid foundation for future academic research, technological development, and industrial applications, driving technological progress and societal development.

### B.2  Limitation

While the CI-GAN model demonstrates significant advancements in Chinese handwriting generation and recognition, some practical limitations could impact its performance in real-world applications. For instance, non-standard or cursive handwriting may pose challenges for accurate signal generation and recognition. Additionally, environmental factors such as external movements or vibrations when using handheld devices could affect the inertial sensor data quality, leading to variations in recognition accuracy. Future work could focus on developing more robust algorithms that account for these real-world variations and improving the model's adaptability to diverse handwriting styles and conditions. These enhancements would ensure that the CI-GAN technology remains effective across a broader range of practical scenarios.

## C  Theory Assumption and Proof

To generate large-scale and high-quality handwriting signals, we introduce optimal transport theory into the generative adversarial network to alleviate mode collapse and mixing issues. We provide a detailed explanation and present a rigorous mathematical proof to show the advantages of this operation.

In traditional conditional GANs, the generator $G$ and the discriminator $D$ are trained by minimizing the loss function $\mathcal{L}_{tradition}$:

$$\mathcal{L}_{tradition} = \min_G \max_D \mathbb{E}_{\mathbf{x} \sim p_{\text{data}}}[\log D(\mathbf{x})] + \mathbb{E}_{\mathbf{z} \sim p_{\mathbf{z}}}[\log(1 - D(G(\mathbf{z})))],$$

where $p_{\text{data}}$ is the real data distribution, and $p_{\mathbf{z}}$ is the distribution of the generator's input noise. This loss function essentially minimizes the Jensen-Shannon Divergence (JSD) between the real data distribution $p_{\text{data}}$ and the generated data distribution $p_g$:

$$\text{JSD}(p_{\text{data}} \| p_g) = \frac{1}{2}\text{KL}(p_{\text{data}} \| M) + \frac{1}{2}\text{KL}(p_g \| M),$$

where $M = \frac{1}{2}(p_{\text{data}} + p_g)$ and KL denotes the Kullback-Leibler divergence. However, JSD has a notable drawback: when the real and generated data distributions do not overlap, the JSD becomes zero, causing the gradients to vanish. This leads to mode collapse, where the generator produces a limited variety of samples.

In optimal transport theory, the Wasserstein distance is utilized to measure the minimum cost of transforming one probability distribution into another. Given two probability distributions $\mu$ and $\nu$ on a metric space $\mathcal{X}$, the Wasserstein distance $W$ is:

$$W(\mu, \nu) = \inf_{\gamma \in \Pi(\mu, \nu)} \mathbb{E}_{(x,y) \sim \gamma}[d(x, y)],$$

where $\Pi(\mu, \nu)$ is the set of all joint distributions whose marginals are $\mu$ and $\nu$, and $d(x, y)$ is a distance metric on $\mathcal{X}$. Therefore, we introduce the Wasserstein distance in optimal transport theory as new loss function $\mathcal{L}_{OT}$, whose objective is to minimize the Wasserstein distance between the generated distribution $p_g$ and the real distribution $p_{\text{data}}$. The $\mathcal{L}_{OT}$ is defined as:

$$\mathcal{L}_{OT} = \min_G \max_{D \in \mathcal{D}} \mathbb{E}_{\mathbf{x} \sim p_{\text{data}}}[D(\mathbf{x})] - \mathbb{E}_{\mathbf{z} \sim p_{\mathbf{z}}}[D(G(\mathbf{z}))]$$

where $\mathcal{D}$ is the set of 1-Lipschitz functions. This Lipschitz constraint can be enforced through weight clipping or gradient penalty. In $\mathcal{L}_{OT}$, the discriminator $D$ is constrained to be 1-Lipschitz:

$$|D(x_1) - D(x_2)| \leq |x_1 - x_2|.$$

This constraint ensures that the discriminator provides meaningful gradients even when $p_g$ and $p_{\text{data}}$ do not overlap. Using the Kantorovich-Rubinstein duality, we can express the Wasserstein distance as:

$$W(p_{\text{data}}, p_g) = \sup_{\|f\|_L \leq 1} \mathbb{E}_{x \sim p_{\text{data}}}[f(x)] - \mathbb{E}_{x \sim p_g}[f(x)].$$

Since $f$ is Lipschitz continuous, it ensures that the gradients $\nabla f(x)$ are bounded and do not vanish. Hence, during the optimization process, the generator receives consistent and informative gradient updates that guide it to produce more realistic and diverse samples. The gradient of the loss function $\mathcal{L}_{OT}$ with respect to the generator's parameters $\theta$ is:

$$\nabla_\theta \mathbb{E}_{\mathbf{z} \sim p_{\mathbf{z}}}[D(G_\theta(\mathbf{z}))] = \mathbb{E}_{\mathbf{z} \sim p_{\mathbf{z}}}[\nabla_\theta D(G_\theta(\mathbf{z}))].$$

This gradient does not vanish even if $p_g$ and $p_{\text{data}}$ have disjoint supports, thanks to the 1-Lipschitz property of $D$. As a result, the generator $G$ can still receive valuable gradient information to adjust its parameters and gradually make $p_g$ approximate $p_{\text{data}}$ even if $p_g$ and $p_{\text{data}}$ do not overlap, effectively addressing mode collapse and mode mixing issues. Overall, after introducing optimal transport theory, we overcome the gradient vanishing problem inherent in traditional GANs, effectively mitigating mode collapse and mode mixing. $\mathcal{L}_{OT}$ maintains the existence and relevance of gradients during training, enabling the generator to continuously improve and produce more diverse and realistic handwriting samples.

