# OpenReview forum: "Chinese Inertial GAN for Writing Signal Generation and Recognition"
_NeurIPS.cc/2024/Conference — Submitted to NeurIPS 2024_

### Official Review · Reviewer_NSpH · 2024-07-04

**Soundness:** 3
**Presentation:** 3
**Contribution:** 3
**Rating:** 6
**Confidence:** 3

**Summary:**

The paper presents a novel Chinese inertial generative adversarial network (CI-GAN) designed to generate high-quality training samples for Chinese writing recognition using inertial sensors. The CI-GAN integrates Chinese Glyph Encoding (CGE), Forced Optimal Transport (FOT), and Semantic Relevance Alignment (SRA) to enhance the quality and authenticity of generated inertial signals. The approach addresses the challenge of collecting diverse and extensive training data for Chinese character recognition, showing significant improvements in classifier performance.

**Strengths:**

The paper introduces innovative methods in the form of CGE, FOT, and SRA, contributing significantly to the field of inertial writing recognition. The release of a new dataset further enriches the community's resources.

**Weaknesses:**

1.Lack of Detailed Baseline Configuration: The paper compares CI-GAN with a traditional GAN in the appendix, but fails to provide detailed settings for the baseline method. This lack of information hinders the ability to fully understand and replicate the comparative effectiveness reported.
2.Insufficient Comparison with Other Augmentation Techniques: The study does not compare CI-GAN with other data augmentation methods, such as random perturbations. It remains unexplored whether applying random disturbances to the data could also substantially improve classifier performance.

**Questions:**

1. Could you provide more information about the baseline model and the CI-GAN model used in your study, such as the number of parameters and other configuration settings?
2. Have you considered evaluating how simple data augmentation methods, such as random perturbations, might also significantly improve classifier performance?

**Limitations:**

The authors have discussed limitations related to the variability of writing styles and the potential impact of environmental factors on sensor data.

---

> ### Author Rebuttal · Authors · 2024-08-06
>
> **For weaknesses 1 and Question 1:**
> The input consists of a 100-dimensional random noise vector and the devised Chinese Glyph Encoding representing the character class, concatenated together to form an input vector. This combined vector passes through a fully connected layer, producing an output of size 256, followed by ReLU activation and batch normalization. The output is then reshaped into a tensor of shape (256, 1), which undergoes a series of 1D transposed convolutional layers. The first transposed convolutional layer uses 512 filters with a kernel size of 7, applying 'same' padding, and is followed by ReLU activation and batch normalization. Subsequent layers reduce the number of filters to 256, 128, 64, and 32, each with a kernel size of 5, maintaining batch normalization and ReLU activation. The final layer consists of 6 filters, corresponding to the six output channels (three for accelerometer and three for gyroscope data).
>
> The input to the discriminator consists of six-channel signals (three for accelerometer data and three for gyroscope data). The signal data undergoes processing through four 1D convolutional layers, which increase in filter count (64, 128, 256, and 512, respectively) and employ kernel sizes of 7 and 5. Each convolutional layer uses 'same' padding and Leaky ReLU activation with an alpha of 0.2, coupled with batch normalization to stabilize the training process. The final convolutional output is flattened and passed through two separate branches. The first branch, designated for real/fake classification, includes a fully connected layer followed by a Sigmoid activation function, yielding a scalar probability that indicates the likelihood of the input being real. The second branch, responsible for class label prediction, also comprises a fully connected layer, followed by a Softmax activation function that outputs a vector of probabilities across all potential classes.
>
> The field of inertial sensors lacks deep learning-based data augmentation methods, and image-based methods are challenging to directly apply to time-series signals, so we chose cGAN as our baseline, which shares the same generator and discriminator as our model. In fact, our CI-GAN only adds three designed modules: CGE, FOT, and SRA. These modules introduce innovative enhancements—CGE provides semantic guidance by encoding the glyph shape of Chinese characters, FOT ensures feature consistency and prevents mode collapse through forced optimal transport, and SRA aligns the semantic relevance between inputs and outputs. These designs result in significant performance improvements, demonstrating the effectiveness and novelty of our approach.
>
> **For Question 2:**
> Following your suggestion, we supplemented extensive comparative experiments, including 12 data augmentation methods covering five major categories. As shown in the Table below (Table 1 in the uploaded PDF), the results clearly demonstrate that CI-GAN significantly outperforms all other methods. Unlike in the field of image processing, it is challenging for humans to recognize semantic information in signal waveforms through observation or to judge whether the augmented signals are reasonable. Therefore, data augmentation methods specifically designed for images are not well-suited for sensor signals. In fact, there is a notable lack of deep learning-based data augmentation methods for inertial sensors. Our CI-GAN fills this gap and has been adopted and applied in the industry.
>
>
>
> | Data Augmentation Methods | 1DCNN | LSTM | Transformer | RF | XGBoost | SVM |
> |---------------------------|-------|------|-------------|----|---------|-----|
> | **Time Domain**           |       |      |             |    |         |     |
> | Cropping                  | 15.7% | 9.1% | 7.7%        | 12.8% | 16.3%  | 9.6% |
> | Noise Injection           | 17.3% | 11.9% | 12.2%      | 8.5% | 13.8%  | 10.1% |
> | Jittering                 | 20.1% | 13.0% | 14.4%      | 9.7% | 17.4%  | 7.5% |
> | **Frequency Domain**      |       |      |             |     |        |     |
> | Amplitude and Phase Perturbations                       | 22.3% | 13.6% | 19.7%      | 19.0% | 25.1%  | 16.3% |
> | Amplitude Adjusted Fourier Transform                     | 32.1% | 20.7% | 25.4%      | 27.5% | 35.9%  | 19.2% |
> | **Decomposition**         |       |      |             |     |        |     |
> | Wavelet                   | 19.9% | 12.1% | 10.6%      | 13.8% | 22.6%  | 9.5% |
> | EMD                       | 24.4% | 17.1% | 20.9%      | 17.9% | 23.4%  | 12.2% |
> | **Mixup**                 |       |      |             |     |        |     |
> | CutMix                    | 21.9% | 14.8% | 15.5%      | 14.7% | 18.9%  | 13.1% |
> | Cutout                    | 25.6% | 16.4% | 16.9%      | 18.5% | 27.1%  | 16.6% |
> | RegMixup                  | 41.5% | 27.8% | 36.8%      | 38.4% | 45.9%  | 30.3% |
> | **Learning based**        |       |      |             |     |        |     |
> | cGAN                      | 18.5% | 14.8% | 15.7%      | 12.4% | 20.5%  | 8.4% |
> | **CI-GAN (ours)**         | **95.7%** | **93.9%** | **98.4%** | **83.5%** | **93.1%** | **74.6%** |
>
>
> We sincerely hope our response addresses your concerns and we will incorporate all your suggested content into the accepted version.

---

> ### Author Response · Authors · 2024-08-12
> **Hoping Our Response Meets Your Expectations**
>
> Thank you once again for your insightful feedback on our paper. We have carefully addressed each of your concerns in our response, particularly regarding the detailed baseline configurations and the comparisons with various data augmentation techniques. We genuinely believe that our work offers valuable advancements in the field, especially given the lack of deep learning-based augmentation methods for inertial sensors.
>
> We would greatly appreciate it if you could take another look at our revisions and explanations. Your feedback is invaluable to us, and we sincerely hope that our paper can meet your expectations and contribute meaningfully to the field.
>
> Thank you for your time and consideration.

---

### Official Review · Reviewer_NmsP · 2024-07-12

**Soundness:** 2
**Presentation:** 3
**Contribution:** 2
**Rating:** 3
**Confidence:** 4

**Summary:**

This paper proposes CI-GAN to acquire unlimited high-quality training samples, alleviating the data scarcity in the inertial signal recognition of Chinese characters. By utilizing these generated data, the performance of recognition models is highly improved.

**Strengths:**

- This paper is easy to follow.
- The proposed methods may help disabled people.

**Weaknesses:**

- The pipeline lacks novelty. The employed technologies are widely used in CV and NLP, and the proposed pipeline merely reuses them for the inertial signal domain without any innovative design. Furthermore, the author fails to cite relevant studies such as [1][2] and does not discuss their differences.
[1] Wasserstein GAN (WGAN)
[2] Efficient Estimation of Word Representations in Vector Space

- The proposed CGE is simply a learnable embedding to represent Chinese characters, lacking innovative design for glyph information. The author introduces GER to enhance the orthogonality of character embeddings but does not provide an ablation study to verify its effectiveness.

- The author uses Wasserstein distance in GANs. What is the difference between this approach and WGAN [1]? Additionally, the author proposes using FFM to supervise the signal in feature spaces. These measures are also similar to some works, such as perceptual loss using VGG and identity loss using ArcFace, but the author does not cite these and discuss the difference.

- The dataset used for training and testing is too small, which could not effectively verify the effectiveness of the proposed method.

**Questions:**

See Weaknesses.

---

> ### Author Rebuttal · Authors · 2024-08-05
>
> **For weakness 1:**
> The novelty of the CI-GAN consists of three proposed modules: Chinese Glyph Encoding (CGE), Forced Optimal Transport (FOT), and Semantic Relevance Alignment (SRA). These modules are interdependent, with many structures serving multiple functions. For example, CGE provides semantic guidance for the generator as a condition and imposes constraints on the generated results within the SRA. Similarly, the signal features in SRA also support the FOT module, making the entire architecture compact and sophisticated.
> Each module also has its innovation. For instance, SRA aligns the relevance between different generation outputs with the relevance between the prompts, significantly reducing hallucinations in the generative model.
> In June 2024, Nature published an article titled "Detecting Hallucination in Large Language Models Using Semantic Entropy," sharing a similar approach to our SRA. They assess the inconsistency in outputs when the same question is posed multiple times to a large model. Their approach essentially forces the model to give similar outputs for similar prompts. Our SRA goes a step further by ensuring that the relationships between the prompts are consistent with the relationships between the outputs, thereby reducing hallucinations and enhancing the model's practicality and stability.
>
> Compared to the CV and NLP fields, the sensor domain lacks deep learning-based data augmentation methods. Most data augmentation techniques for images are not applicable to time-series signals. Our method is pioneering in the inertial sensor domain and has been adopted by a wearable device manufacturer. We apologize for missing some literature, and we will reference the recommended papers and similar work published in Nature in future versions.
>
> **For weakness 2:**
> Unlike general embeddings that capture character meaning, CGE represents character glyph. Inertial sensors capture writing motions containing glyph information, allowing us to construct glyph features of each character during training. To achieve this, we introduce an encoding matrix following the one-hot input and design a Glyph Encoding Regularization (GER) based on Rényi entropy. As GER decreases during training, the Rényi entropy of the glyph encoding matrix increases, leading to two key effects: Each vector's information entropy increases, enabling it to carry more glyph information; The orthogonality between different encodings improves, capturing the differences between glyphs. This design may also benefit other categorical representation tasks by applying a similar regularization term to the category encoding dictionary. Additionally, in CI-GAN, CGE supports the SRA module, helping to alleviate hallucinations in the generative model. Moreover, CGE is also used in FFM to ensure consistency between real signal features, generated signal features, and glyph encoding, where CGE is also directly supervised.
>
> The effectiveness of CGE is primarily due to GER, which is why we provided an ablation study of CGE without showcasing the ablation results of GER separately. Following your suggestion, we removed GER while retaining the encoding matrix, reducing it to a learnable transformation from one-hot encoding without additional guidance. This led to a significant performance decline, as shown in Table 2 (in the uploaded PDF), underscoring the critical importance of the Rényi entropy-based regularization in categorical representation tasks.
>
> **For weakness 3:**
> WGAN primarily addresses the overall distribution differences between generated and real samples. In contrast, FOT incorporates a Forced Feature Matching (FFM) mechanism that enhances the realism of the generated signals by aligning their features with those of real samples. Unlike WGAN, FFM imposes an additional constraint, ensuring that the generated samples not only match the real data distribution closely but also maintain consistency in key features. This feature-specific matching is not explicitly achieved in WGAN, which is crucial for signal generation. Unlike images, the quality of signals cannot be readily assessed through visual inspection. Thus, stringent constraints are essential to ensure the reliability of the generated results.
>
> Perceptual loss only constrains the consistency between generated and real signals. Identity loss in ArcFace ensures that the generated face images retain the same identity as the real faces. Differently, FFM proposes triple consistency constraints for generative models: prompt, generated signal features, and true signal features, which not only improves the realism of the generated signals but also ensures their SEMANTIC accuracy. Meanwhile, FFM also supervises the glyph encoding, reflecting the interaction between the proposed three modules. We will supplement these analyses and citations in future versions.
>
> **For weakness 4:**
> We recruited two new volunteers, each using their smartphone to write 500 characters in the air, resulting in 1000 new samples. Writing Chinese characters, segmenting, and extracting corresponding signals for each character is extremely labor-intensive. Particularly, the segmentation phase requires optical equipment to precisely mark the start and end times of the signal segments corresponding to each character, which is a highly time-consuming task.
>
> Given the excellent performance of CI-GAN, we used all 1000 newly collected samples for testing without retraining. As shown in Table 3 (in the uploaded PDF), the six classifiers performed even better, whose improvement is likely because these smartphones were newer, and the built-in sensors had not aged much, resulting in higher-quality signals. This demonstrates that the classifiers trained with CI-GAN-generated signals can adapt to sensors of different usage times, providing significant convenience for device manufacturers and leading to CI-GAN adoption and application in the industry.
>
> Finally, we sincerely hope to receive your recognition.

---

> ### Author Response · Authors · 2024-08-12
> **Hoping Our Response Meets Your Expectations**
>
> Thank you once again for your thoughtful review and valuable feedback on our paper. We have taken your comments very seriously and have provided detailed explanations and clarifications in our response. We understand that assessing innovation can sometimes involve different perspectives, and we sincerely hope that you will consider our explanations, particularly regarding the aspects of novelty that we have highlighted.
>
> Our research not only introduces some unique design elements but has also shown promising results in practical applications. We genuinely believe that this work can make a meaningful contribution to both the academic community and industry. If there are any further questions or areas that need clarification, we would be more than happy to discuss them in greater detail.
>
> We truly appreciate your time and effort and hope that our work can meet your expectations.

---

> > ### Comment · Reviewer_NmsP · 2024-08-13
> >
> > Thank you for your detailed rebuttal and the additional explanations provided. After carefully considering your responses, I still have concerns preventing me from recommending acceptance.
> >
> > - Lack of Clear Innovation: The CI-GAN framework seems to be a variation of conditional GAN, and the CGE appears to be an upscaled one-hot embedding rather than a novel integration of glyph information. Without clear motivation and experimental validation, the proposed modules show differences but not true innovation. The concept of hallucination introduced in the rebuttal also seems unrelated to the core task, making it difficult to understand its relevance.
> >
> > - Overcomplication with Intuitive Modules: The paper introduces several modules based on intuitive motivations, which makes it hard to identify a central, innovative contribution. The work feels more like a collection of empirical studies than a focused research effort.
> >
> > - Insufficient Experimental Validation: I still find the experiments lacking. For example, there is no visualization of the generated handwritten characters or an assessment of their diversity, which would be critical in evaluating the effectiveness of CI-GAN.
> >
> > I hope these points help you refine your work in the future.

---

> > > ### Author Response · Authors · 2024-08-14
> > >
> > > Thank you for providing additional feedback. We would like to address the remaining concerns you have raised.
> > >
> > > 1. Lack of Clear Innovation:
> > > We understand that you perceive CI-GAN as a variation of c-GAN, but if we follow this logic, c-GAN itself could also be considered a variation of GAN. By this reasoning, all models based on the GAN architecture could be seen as mere variations, thereby discounting any innovations in the field. However, we strongly believe that the true measure of innovation lies in the specific adaptations and design made to the base architecture to address unique challenges—in our case, the challenges inherent to inertial signal generation.
> > >
> > > Regarding the Chinese Glyph Encoding (CGE), it is indeed more than just an “upscaled one-hot embedding.” CGE was designed as a novel way of category encoding that injects more information into the generative process. We devised a Renyi entropy-based regularization applied to a learnable category encoding matrix, significantly enhancing the matrix's ability to represent categorical information. This allows CGE to capture the nuances of Chinese character glyphs, which is a novel approach in the context of signal generation. To support this, we have visualized the glyph encodings of various Chinese characters, demonstrating that characters with similar glyphs are indeed closer in the embedding space. This visualization underscores the effectiveness of CGE in preserving the structural relationships between different characters.
> > >
> > > Additionally, we are puzzled by your comment that “The concept of hallucination introduced in the rebuttal also seems unrelated to the core task.” Hallucination is a well-known issue in generative models, particularly in scenarios where the generated outputs can deviate from realistic or expected patterns. Our SRA module is specifically designed to mitigate hallucination by ensuring that the semantic relationships between generated signals are consistent with those of the input glyphs, thereby enhancing the realism and reliability of the generated signals. Addressing hallucination is not only relevant but crucial to the success of our generative task.
> > >
> > > 2. Value of Intuitive Motivations:
> > > We respectfully disagree with the notion that intuitive design somehow detracts from the innovation or significance of our contributions. On the contrary, intuitive designs often lead to more effective and impactful solutions precisely because they resonate with the underlying principles of the problem being addressed. Each of the three modules in CI-GAN—CGE, FOT, and SRA—was designed with clear, intuitive motivations, and each one independently offers significant contributions to the field of deep learning. For example:
> > >
> > > CGE introduces a novel way of category encoding that injects more information into the generative process, enhancing the quality and diversity of the generated signals.
> > > FOT establishes a triplet constraint between the input, output, and label, addressing issues such as mode collapse, mode mixing, and the authenticity of generated results. This approach helps to ensure that the generated signals are not only diverse but also semantically accurate and realistic.
> > > SRA ensures that the semantic relationships between inputs are maintained in the generated outputs, reducing the likelihood of hallucinations.
> > > These modules are not merely empirical tweaks; they represent fundamental advancements that can inspire future research. When integrated into the CI-GAN framework, these modules work synergistically to achieve the first successful generation of inertial sensor signals. This is not just a collection of empirical studies; it is a cohesive and innovative solution to a complex problem that has not been addressed before.
> > >
> > > 3. Visualization and Diversity:
> > > Your comment that “there is no visualization of the generated handwritten characters” seems to overlook the visualizations we provided in the paper. Figure 3 illustrates the generated signals for different Chinese characters, showing how they closely follow the fluctuation trends of real signals. To further emphasize diversity, Figure 4 presents the results for multiple generated signals of the same character, “王,” compared with a real handwriting signal. These visualizations clearly demonstrate that CI-GAN not only generates diverse samples but also maintains the essential characteristics of real handwriting signals. The differences in individual features, while preserving overall trends, validate the model’s ability to produce high-quality, diverse, and realistic samples.
> > >
> > > We hope this response clarifies the innovations and contributions of our work. We believe CI-GAN represents a significant advancement in the field of inertial signal generation and has the potential to drive future research and applications. We sincerely appreciate your thoughtful consideration of our work and hope that you will recognize the value it brings.
> > >
> > > Thank you for your time and consideration.

---

### Official Review · Reviewer_3apC · 2024-07-15

**Soundness:** 2
**Presentation:** 3
**Contribution:** 2
**Rating:** 5
**Confidence:** 2

**Summary:**

The paper address an important probem in human computer interaction: making computers accessible to vision impaired people. The paper address this my collection paired data of text and imu signals. First, the paper address the issues of limited data by training a generative model, to resample/bootstrap more data and then train recognition model on both real and generated data to archive high performance.

**Strengths:**

the paper addresses an important social problem, and accessibility should be focused on all groups.

The data collected for this paper, the paired data on text and imu is very useful, hope the authors will open-source it.

paper is well written and the figures are clear and convey the ideas.

**Weaknesses:**

My main concern is, that it is very unlikely that we get more than we give to the system, the generated samples are a function of real samples.
I would like to see, a competitive baseline with good data augmentation, and maybe on a low data regime gan generated samples are better than augmentation, but this has to be shown, otherwise, I don't see the value of extra effort to train a generative model to get data augmentation.

**Questions:**

please see my concerns about the weakness section.

**Limitations:**

I wouldn't say this is a major limitation, but on the scale axis, this problem can be solved by collecting more data. Unlike annotations like explaining an image or video, handwriting signals are more easy to collect on the long term. would be nice if the authors can address this, also please explain the issues with data augmentation.

---

> ### Author Rebuttal · Authors · 2024-08-05
>
> We fully agree with your point: "It is very unlikely that we get more than we give to the system." In fact, what we give to the system is sufficient, as our training data provides multiple writing signals for each Chinese character. In comparison, humans can usually recognize new categories after just one exposure. This suggests that the model already receives enough information; the key is to utilize this information to thoroughly explore and memorize the intrinsic patterns of each Chinese character and then generate reasonable variations.
> To memorize the patterns of each character, we designed the Chinese Glyph Encoding (CGE) module, which effectively represents the shapes and stroke features of Chinese characters, providing a solid informational foundation for generating new writing signals.
> To generate reasonable variations, we introduced the Forced Optimal Transport (FOT) and Semantic Relevance Alignment (SRA) mechanisms. FOT addresses common issues in GAN, such as mode collapse and mixing, and establishes a triple constraint involving the prompt, generated signal features, and real signal features, ensuring the generated signals' authenticity and semantic accuracy. The SRA mechanism aligns the relationships of generated signals with the relationships of input Chinese glyph encodings, providing a group-level constraint, which mitigates hallucinations in the generative model, resulting in more realistic and reliable signal samples.
>
>
> Following your recommendation, we employed five major categories of data augmentation—Time Domain, Frequency Domain, Decomposition, Mixup, and Learning-based strategies—encompassing 12 methods for comparison. All methods generated the same amount of samples (15,000) for training six classifiers, as shown in Table below (Table 1 in the uploaded PDF). Notably, except for our proposed augmentation method, the accuracy of classifiers trained using all other data augmentation methods failed to surpass 50%, whereas our method achieved over 90%. Additionally, due to the lack of deep learning-based augmentation methods in the sensor field, we could only compare our approach with cGAN, which performed worse than many non-deep learning methods, underlining the difficulty of designing deep learning models capable of generating accurate and realistic inertial handwriting signals and highlights the value of our CI-GAN.
>
>
>
> | Data Augmentation Methods | 1DCNN | LSTM | Transformer | RF | XGBoost | SVM |
> |---------------------------|-------|------|-------------|----|---------|-----|
> | **Time Domain**           |       |      |             |    |         |     |
> | Cropping                  | 15.7% | 9.1% | 7.7%        | 12.8% | 16.3%  | 9.6% |
> | Noise Injection           | 17.3% | 11.9% | 12.2%      | 8.5% | 13.8%  | 10.1% |
> | Jittering                 | 20.1% | 13.0% | 14.4%      | 9.7% | 17.4%  | 7.5% |
> | **Frequency Domain**      |       |      |             |     |        |     |
> | Amplitude and Phase Perturbations                       | 22.3% | 13.6% | 19.7%      | 19.0% | 25.1%  | 16.3% |
> | Amplitude Adjusted Fourier Transform                     | 32.1% | 20.7% | 25.4%      | 27.5% | 35.9%  | 19.2% |
> | **Decomposition**         |       |      |             |     |        |     |
> | Wavelet                   | 19.9% | 12.1% | 10.6%      | 13.8% | 22.6%  | 9.5% |
> | EMD                       | 24.4% | 17.1% | 20.9%      | 17.9% | 23.4%  | 12.2% |
> | **Mixup**                 |       |      |             |     |        |     |
> | CutMix                    | 21.9% | 14.8% | 15.5%      | 14.7% | 18.9%  | 13.1% |
> | Cutout                    | 25.6% | 16.4% | 16.9%      | 18.5% | 27.1%  | 16.6% |
> | RegMixup                  | 41.5% | 27.8% | 36.8%      | 38.4% | 45.9%  | 30.3% |
> | **Learning based**        |       |      |             |     |        |     |
> | cGAN                      | 18.5% | 14.8% | 15.7%      | 12.4% | 20.5%  | 8.4% |
> | **CI-GAN (ours)**         | **95.7%** | **93.9%** | **98.4%** | **83.5%** | **93.1%** | **74.6%** |
>
>
>
> In practice, collecting, segmenting, and processing these handwriting signals is a challenging task. We need to isolate and extract the segment corresponding to each character from a continuous signal flow, a process that requires identifying each character's precise start and end points, as shown in Figure 1 in the uploaded PDF. These points are not easily identifiable and often require optical equipment for accurate frame-level segmentation and annotation. We invested significant time and effort to obtain the original 4,500 handwriting signals. Our CI-GAN eliminates these difficulties by providing a straightforward method for generating handwriting signals, thereby saving time and resources.
>
> Thank you for your insightful comment. We hope we have addressed your concerns.

---

> ### Author Response · Authors · 2024-08-12
> **Hoping Our Response Meets Your Expectations**
>
> I hope this message finds you well. I wanted to take a moment to sincerely thank you for your thoughtful and detailed review of our submission. Your insights have been invaluable in helping us refine our work, and we have made considerable efforts to address each of your concerns thoroughly.
>
> In our response, we provided a detailed comparison with various data augmentation methods to highlight the significant value of our CI-GAN approach, which introduces generative deep learning models into inertial sensor data augmentation for the first time. Our results, which include extensive testing and ablation studies, show that CI-GAN not only performs significantly better than other methods but also offers a flexible platform that addresses the specific challenges of the sensor signal domain. We genuinely believe that our approach brings innovation to this field, particularly through the designed Chinese Glyph Encoding, Forced Optimal Transport, and Semantic Relevance Alignment, which together form a cohesive and effective system. These elements work in tandem to ensure not just the generation of realistic signals but also their alignment with the semantic content, which is crucial for practical applications.
>
> Thank you once again for your time and effort. We are eagerly looking forward to your response.

---

### Author Rebuttal · Authors · 2024-08-05

We sincerely thank the reviewers and the conference chair for their valuable feedback and thoughtful consideration of our paper. First, we want to clarify that collecting handwriting samples of Chinese characters is not easy. During data collection, volunteers wrote different Chinese characters continuously. We had to accurately locate the signal segments corresponding to each character from long signal streams, as shown in Figure 1 in the uploaded PDF.
However, accurately segmenting and extracting signal segments requires synchronizing optical motion capture equipment and then comparing the inertial signals frame by frame with the optical capture results to find all character signal segments' starting and ending frames. Consequently, we expended significant time and effort to obtain 4,500 signal samples in this paper, establishing the first Chinese handwriting recognition dataset based on inertial sensors, which we have made open-source partially. By contrast, our CI-GAN can directly generate handwriting motion signals according to the input Chinese character, eliminating the complex processes of signal segmentation, extraction, and cleaning, as well as the reliance on optical equipment. We believe it provides an efficient experimental data platform for the field.

Unlike the fields of CV and NLP, many deep learning methods have not yet been applied to the sensor domain. More importantly, unlike image generation, where the performance can be visually judged, it is challenging to identify semantics in waveforms by observation and determine whether the generated signal fluctuations are reasonable, which imposes high requirements on generative model design. Therefore, we had to design multiple guidance and constraints for the generator, resulting in the design of Chinese Glyph Encoding (CGE), Forced Optimal Transport (FOT), and Semantic Relevance Alignment (SRA).

* CGE introduces a regularization term based on Rényi entropy, which increases the information content of the encoding matrix and the distinctiveness of class encodings, providing a new category representation method that can also be applied to other tasks. As far as we know, this is the first embedding targeted at the shape of Chinese characters rather than their meanings, providing rich semantic guidance for generating handwriting signals.
* FOT establishes a triple-consistency constraint between the input prompt, output signal features, and real signal features, ensuring the authenticity and semantic accuracy of the generated signals and preventing mode collapse and mixing.
* SRA constrains the consistency between the semantic relationships among multiple outputs and the corresponding input prompts, ensuring that similar inputs correspond to similar outputs (and vice versa), significantly alleviating the hallucination problem of generative models. Notably, the June 2024 Nature paper "Detecting Hallucination in Large Language Models Using Semantic Entropy," published after our NeurIPS submission, shares a similar idea with our proposed SRA. They assess model hallucination by repeatedly inputting the same prompts into generative models and evaluating the consistency of the outputs. Their approach essentially forces the model to produce similar outputs for similar prompts. Our SRA not only achieves this but also ensures that the relationships between prompts are mirrored in the relationships between the outputs. This significantly reduces hallucinations and enhances the model's practicality and stability.

CGE, FOT, and SRA not only guide and constrain the generator but also interact with each other. We added a diagram (Figure 2 in the uploaded PDF) to illustrate their roles and interactions. The Chinese glyph encoding not only provides semantic guidance to the generator but also supplies the necessary encoding for FOT and SRA, and it is also supervised in the process. FOT and SRA share the VAE and generated signal features, providing different constraints for the generator, with FOT focusing on improving signal authenticity and enhancing the model's cognition of different categories through the semantic information injected by CGE, thereby mitigating mode collapse and mode mixing. In contrast, SRA ensures consistency between the relationships of multiple outputs and prompts through group-level supervision, which helps alleviate the hallucination problem of generative models.

In summary, the three modules proposed in CI-GAN, CGE, FOT, and SRA are innovative and interlinked, significantly enhancing the performance of GANs in generating inertial sensor signals, as evidenced by numerous comparative and ablation experiments. This method is a typical example of deep learning empowering the sensor domain and has been recognized by the industry and adopted by a medical wearable device manufacturer. It has the potential to become a benchmark for data augmentation in the sensor signal processing field. We sincerely hope we have addressed the concerns of the three reviewers, and once again, we thank everyone for their review and suggestions for this paper.

---

### Author Response · Authors · 2024-08-13

I hope this message finds you well. I apologize for reaching out again, but with only one day left in the discussion period and no responses yet, I am feeling anxious about the status of our submission. This paper is of immense importance to us, and we have put in a great deal of effort to thoroughly address every concern and suggestion raised in your initial reviews.

We genuinely believe that the experimental results and detailed explanations provided in our response have resolved the issues you highlighted. This work is critical not only to our research but also has the potential to make a significant impact in the field.

I humbly and earnestly request that you please review our responses and provide your final feedback. Your input is invaluable, and I sincerely hope that our efforts meet your expectations.

Thank you so much for your time and understanding.

---

### Author Response · Authors · 2024-08-13

I apologize for reaching out once more, but with less than 24 hours remaining in the discussion period, I am deeply concerned as we have not yet received any responses to our detailed rebuttals. This paper is incredibly important to us, both for our ongoing research and for its potential contributions to the field.

We have worked tirelessly to address every concern and suggestion raised in your initial reviews, providing thorough explanations and supplementing a lot of experiments in 7 days. We genuinely believe that our work brings valuable innovations that could greatly benefit the research community. It would be truly disheartening for this paper to go unnoticed or unacknowledged.

With the utmost respect and sincerity, I humbly request that you please review our responses and provide your final feedback before the discussion period ends. Your input is crucial to us, and we are deeply hopeful that our efforts will meet your expectations.

Thank you for your time, consideration, and understanding during this critical time.

---

### Decision · Program_Chairs · 2024-09-25

**Decision:**

Reject

**Comment:**

The paper proposes CI-GAN, a generative adversarial network designed to produce high-quality inertial sensor data for Chinese character recognition, aiming to improve accessibility for vision-impaired individuals. Reviewers (R-3apC, R-NmsP, R-NSpH) acknowledge the significance of addressing this social problem and the potential benefits. However, concerns are raised about the lack of novelty in methodology and insufficient comparisons with standard data augmentation techniques (R-NmsP, R-3apC, R-NSpH). Additionally, the small dataset size (R-NmsP) and lack of detailed baseline configurations (R-NSpH) limit the validation of the proposed approach. Although the reviewers didn't participate in the discussion phase (after being contacted by the AC), the AC feels the paper is not ready for publication.